# Genome-Wide Identification of Callose Synthase Family Genes and Their Expression Analysis in Floral Bud Development and Hormonal Responses in *Prunus mume*

**DOI:** 10.3390/plants12244159

**Published:** 2023-12-14

**Authors:** Man Zhang, Wenhui Cheng, Jia Wang, Tangren Cheng, Xinlian Lin, Qixiang Zhang, Cuiling Li

**Affiliations:** 1Beijing Key Laboratory of Ornamental Plants Germplasm Innovation & Molecular Breeding, National Engineering Research Center for Floriculture, Beijing Laboratory of Urban and Rural Ecological Environment, Engineering Research Center of Landscape Environment of Ministry of Education, Key Laboratory of Genetics and Breeding in Forest Trees and Ornamental Plants of Ministry of Education, School of Landscape Architecture, Beijing Forestry University, Beijing 100083, China; manzhang_bjfu_edu@163.com (M.Z.); cwh13452454084@bjfu.edu.cn (W.C.); wangjia8248@163.com (J.W.); chengtangren@163.com (T.C.); 2Flower Research Institute, Meizhou Academy of Agriculture and Forestry Sciences, Meizhou 514071, China; wanglovesky@163.com

**Keywords:** callose synthase, *Prunus mume*, gene family analysis, evolutionary analysis, expression analysis

## Abstract

Callose is an important polysaccharide composed of beta-1,3-glucans and is widely implicated in plant development and defense responses. Callose synthesis is mainly catalyzed by a family of callose synthases, also known as glucan synthase-like (GSL) enzymes. Despite the fact that GSL family genes were studied in a few plant species, their functional roles have not been fully understood in woody perennials. In this study, we identified total of 84 *GSL* genes in seven plant species and classified them into six phylogenetic clades. An evolutionary analysis revealed different modes of duplication driving the expansion of GSL family genes in monocot and dicot species, with strong purifying selection constraining the protein evolution. We further examined the gene structure, protein sequences, and physiochemical properties of 11 GSL enzymes in *Prunus mume* and observed strong sequence conservation within the functional domain of PmGSL proteins. However, the exon–intron distribution and protein motif composition are less conservative among *PmGSL* genes. With a promoter analysis, we detected abundant hormonal responsive cis-acting elements and we inferred the putative transcription factors regulating *PmGSLs*. To further understand the function of GSL family genes, we analyzed their expression patterns across different tissues, and during the process of floral bud development, pathogen infection, and hormonal responses in *Prunus* species and identified multiple GSL gene members possibly implicated in the callose deposition associated with bud dormancy cycling, pathogen infection, and hormone signaling. In summary, our study provides a comprehensive understanding of GSL family genes in *Prunus* species and has laid the foundation for future functional research of callose synthase genes in perennial trees.

## 1. Introduction

Callose, also known as β-1,3-glucan, is a type of glucan polymers connected by β-1,3-glycosidic bonds. Callose is widespread in higher plants and plays an essential part in plant growth and development [1]. Normally, callose accumulates in cell walls or cell wall-related structures, such as cell plates, outer cell walls of pollen, and plasmodesmata. During cytokinesis, callose accumulates on the surface of cell plates, which is important for septum formation [2]. Callose is also required for pollen grain formation and pollen tube wall assembly [3,4]. Additionally, callose is found localized in phloem sieve plates and at the cell plasmodesmata to regulate the permeability of phloem and the intercellular transport of water, nutrients, and signal molecules [5]. Under stress conditions such as wounding or pathogen infection, callose quickly accumulates, blocking the sieve pores and plasmodesmata, thereby acting as physical barrier to prevent the penetration of the pathogen and seal injured tissues [6].

Callose synthases (CALSs), also referred as glucan synthase-like (GSL) enzymes, are the key enzymes catalyzing callose synthesis using UDP-glucose as initial substrate. The GSL proteins are often located at the plasma membrane and contain multiple transmembrane segments and a hydrophilic central loop, which is the main site for callose synthesis [7]. GSL proteins often function in the form of multi-subunit enzyme complexes by interacting with phragmoplastin, sucrose synthase (SuSy) [8], and UDP-glucose transferase (UGT1) [7]. The Rop1 protein (homologue of Rho protein) binds to glucosyltransferase through protein–protein interaction and regulates the β-1,3-glucan synthase in yeast and the callose synthase in plants [7].

In the past decades, GSL family genes have been identified and functionally studied in a few plant species. In *Arabidopsis*, twelve callose synthase genes were identified and designated as *AtCALS1*-*AtCALS12* [9] or *AtGSL1*-*AtGSL12* based on two independent nomenclature systems [10]. The more frequently used is the gene name GSL derived from ’glucan synthase-like’ [11]. The phylogenetic analysis of *AtGSLs* suggested four subfamilies, members of which exert similar functional roles [11]. For example, *AtGSL1* and *AtGSL5* within clade 1 function redundantly in pollen development and male sterility [12]. Similarly, the rice homologue *GSL5* was found to be essential for microspore development and male fertility [13]. *AtGSL8* and *AtGSL10* were found to be responsible for male gametophyte development and plant growth. Arabidopsis *gsl8* and *gsl10* mutants display a dwarfed growth habitat, asymmetric microspore division, and failed entry of microspore into mitosis [14]. *AtGSL8* has also been reported to regulate the plasmodesmata permeability and tropic response. Mutations in *AtGSL8* lead to reduced callose deposition at plasmodesmata and increased SEL (size exclusion limit) in leaf epidermal cells [15]. Moreover, *AtGSL7* regulates the callose deposition at plasmodesmata during early development of phloem sieve plates and is induced by wounding in mature phloem [5]. In addition, *GSL* genes were found to be involved in plant defense responses against pathogen infection, such as *AtCALS1* (*AtGSL6*), *AtCALS8* (*AtGSL4*), and *AtCALS12* (*AtGSL5*) [16,17]. With the application of salicylic acid (SA), *AtGSL1* and *AtGSL5* were strongly induced in *Arabidopsis* leaves, displaying a similar expression pattern as after exposure to *Hyaloperonospora* infection [17]. *AtGSL5* was reported to be responsible for callose synthesis in sporophytic tissue in response to wounding or pathogen infection. Mutations in *GSL5* resulted in depletion of callose in papillae and more resistance against pathogens [18]. In cotton, the overexpression of a callose synthase gene *GhCalS5* significantly enhanced the callose formation and increased cotton resistance against cotton aphids [19].

Despite the in-depth understanding of glucan synthase genes in *Arabidopsis* and crops, previous studies regarding to the functional role of *GSLs* in woody perennials are still limited. In pear (*Pyrus bretschneideri*), *PbrCalS5* was specifically highly expressed in pollen tubes. The knockdown of *PbrCalS5* leads to decreased callose deposition in pollen tube walls and subsequent inhibition of pollen tube growth, indicating its role in pear pollen tube growth [20]. On the other hand, the *GSLs* also participate in plant adaptive responses to environmental cues [6]. Under short day conditions, high levels of ABA trigger the expression of *callose synthase 1*, which results in increased callose synthesis and deposition at plasmodesmata in poplar shoot tips [21,22]. The blocked plasmodesmata prevent growth-promoting signals from accessing the shoot meristems and ensure the arrested growth during the dormancy period throughout cold winters [22].

*Prunus* is among the most important genera in the *Rosaceae* family due to its economic value as common fruit and for its ornamental value [23]. Within this genus, *Prunus mume* is recognized as the most early flowering tree species, which can bloom under low temperatures in early spring [24]. The flower buds of *Prunus mume* initiate in late summer, followed by leaf drop and establishment of dormancy triggered by short daylength and low temperature. After exposure to sufficient chilling, the floral buds resume growth and bloom under suitable environmental conditions [25]. In the context of global climate change, the elevated temperatures have put threats to many temperate tree species by disrupting their cold acclimation progression and dormancy cycling [26]. Therefore, it is important to fully dissect the gene regulatory networks underlying dormancy cycling in woody perennials. Though dynamic callose deposition was observed in a few perennial species during bud dormancy cycling process, the functional role of *GSLs* has not been systematically analyzed in *Prunus* species yet. In this study, we first identified the *GSL* gene members and analyzed their gene evolution and expansion mechanism across seven plant species. Furthermore, we focused on analyzing the transcriptional regulation and expression profiles of *PmGSL* genes during bud dormancy cycling and hormonal responses in *Prunus mume*. Our work has provided theoretical understanding for the functional role and transcriptional regulation of glucan synthase-like genes in tree species.

## 2. Results

### 2.1. The Identification of GSL Family Genes across Seven Plant Species

To identify GSL family genes in seven plant species, we employed HMM search and BLASTP to search for putative gene members. After combining the results of the two approaches, we identified a total of 72 genes, including 11 genes in *Oryza sativa*, 12 genes in *Sorghum bicolor*, 11 genes in *Populus tricocarpa*, 18 genes in *Malus domestica*, 11 genes in *Prunus mume*, and 9 genes in *Prunus persica* (Appendix A). The GSL family proteins were further validated to contain the GSL domain using NCBI CDD search and were blasted against the NCBI non-redundant protein database (Appendix A). The GSL family members were renamed by their sequential order across chromosomes for each species (Appendix A). The length and molecular weight of GSL proteins varied greatly among different species. In *Arabidopsis*, GSL proteins were of similar length (1768 aa to 1976 aa) with the molecular weight ranging from 205.52 to 228.48 kDa, while GSL paralogous proteins from *O. sativa*, *M. domestica,* and *S. bicolor* differed greatly within each species. For example, the length of GSL proteins ranged from 469 to 1919 aa and the molecular weight ranged from 49.55 to 218.92 kDa in rice (Appendix A).

### 2.2. Phylogenetic Analysis of GSL Family Genes

To study the phylogenetic relationship among GSL family genes, we generated a phylogenetic tree for GSL proteins from *A. thaliana*, *O. sativa*, *S. bicolor*, *Populus tricocarpa*, *M. domestica*, *P. mume,* and *P. persica* using the maximum likelihood algorithm. (Figure 1). The 84 GSL family proteins can be split into six major clades with the gene numbers unevenly distributed across groups (Figure 1). Clade I contained the maximum number of 19 gene members that were clustered with AtGSL6, AtGSL3, AtGSL12, and AtGSL9. Clade III and Clade IV were the two smallest groups of GSL proteins, which were grouped with their *Arabidopsis* homologues, AtGSL4 and AtGSL2, respectively. Within Clade I, PmGSL9, PmGSL10, and PmGSL11 were clustered with AtGSL6, AtGSL3, AtGSL12, and AtGSL9. PmGSL7 and two *Arabidopsis* GSLs, AtGSL7 and AtGSL11 were grouped into Clade II. PmGSL8, PmGSL1, and PmGSL2 fell into Clade VI with their corresponding *Arabidopsis* orthologs AtGSL8 and AtGSL10. Clade V, on the other hand, contained 13 GSL proteins that were closely related to AtGSL1 and AtGSL5 (Figure 1). Among each clade, orthologous GSL proteins from *O. sativa* and *S. bicolor* were closely related, forming subgroups separate from the GSL proteins from the other five species (Figure 1). For dicot species, the GSLs from *P. mume* were first grouped with those of *P. persica* and *M. domestica*, then with the gene cluster of *Populus tricocarpa* and *Arabidopsis* (Figure 1). We identified the largest number of *GSL* genes in *M. domestica* within each clade, which was almost twice of that in *Arabidopsis*.

### 2.3. Protein Physiochemical Properties of PmGSL Proteins

The PmGSL proteins ranged from 761 to 1970 amino acids with an average length of 1645 aa (Table 1). The molecular weight of PmGSL proteins was between 85,842.58 and 227,615.31 Da. We also assessed the isoelectric point (PI), instability index, and the grand average of hydropathy (GRAVY) of PmGSL proteins. All PmGSL proteins shared high isoelectric point (pI) values > 8.0 (Table 1). The predicted instability indexes of PmGSL2, PmGSL3, PmGSL4, PmGSL5, and PmGSL9 are higher than 40 (Table 1). The aliphatic indexes among PmGSLs were between 89.11 and 100.7 (Table 1). The predicted grand average of hydropathicity (GRAVY) values suggested that most PmGSL proteins are hydrophilic except for PmGSL1 and PmGSL3 (Table 1). The transmembrane helix number of PmGSL proteins ranged from 8 to 16. According to the subcellular localization prediction, PmGSL proteins are all located in the plasma membrane (Table 1).

### 2.4. Gene Structure and Motif Analysis of PmGSLs

We investigated the phylogenetic relationship among PmGSL proteins and observed a slightly different subcluster structure compared to that of all GSL proteins (Figure 2a). The protein motif analysis revealed five featured motifs, which were universally present except for PmGSL1 and PmGSL2 (Figure 2a; Appendix A). PmGSL1 was the only protein lacking motif 1, while PmGSL2 only contained motif 1 (Figure 2b). According to the Conserved Domain Database, all PmGSL proteins contained a complete featured glucan synthase domain (Figure 2b). Additionally, most PmGSLs contained the FKS1 domain, which encodes the alternative catalytic subunits of glucan synthases, except for PmGSL10 and PmGSL1. We also detected the Vita1 (VPS20-associated protein 1) domain within PmGSL11, PmGSL9, PmGSL5, PmGSL4, and PmGSL7 (Figure 2b). By analyzing the protein sequence alignment of GSL functional domains, we observed relatively conserved residues within the region of GSL domains among PmGSL proteins (Appendix A). The *PmGSL* genes displayed a less conserved gene structure (Figure 2c). *PmGSL3* and *PmGSL6* only have one exon, while the rest of the *PmGSL* family genes have 19 to 47 exons (Figure 2c). The chromosome locations of GSL family genes were studied based on the gene annotation of the *P. mume* genome (Appendix A). The 11 *PmGSL* genes were scattered unevenly across eight chromosomes in *P. mume*. Chromosome 3 contained three *PmGSL* genes (*PmGSL3*, *PmGSL4*, and *PmGSL5*), while chromosome 1 and 4 contained two copies of *GSL* genes (Appendix A). The rest of the *PmGSLs* were anchored on three scaffolds in the form of a single gene copy (Appendix A).

### 2.5. Evolutionary Analysis of GSL Family Genes

To understand the evolutionary relationship among GSL family genes, we performed a collinearity analysis among *GSL* genes from seven species. The results showed extensive genome synteny among dicot species including *P. persica*, *P. mume*, *M. domestica, Populus tricocarpa,* and *Arabidopsis*, and between the two monocots *O. sativa* and *S. bicolor* (Figure 3). We detected 7 *PmGSL* genes in collinearity with 8 *PpGSLs*, which are in synteny with 13 *GSL* genes in *M. domestica* (Figure 3). We also identified 10 *MdGSL* genes, which are orthologous to 7 *GSLs* in *Populus tricocarpa*. Despite extensive genome synteny between *A. thaliana* and *Populus tricocarpa*, only two *GSLs* were found to be collinear between these two species (Figure 3). For monocot species, the genomes of *O. sativa* and *S. bicolor* have 8 *GSL* genes in collinearity. Interestingly, no collinear *GSL* orthologs were found between *A. thaliana* and *O. sativa*, indicating the dicot and monocot species may have undergone lineage-specific chromosome re-organization after speciation.

Gene duplication acts as one of the major forces driving the evolution of gene families [27]. Therefore, we also analyzed the expansion mechanism of GSL family genes among the seven investigated species (Appendix A). Dispersed duplication, WGD/segmental duplication, and tandem duplication are three main duplication modes detected among *GSL* genes of most dicotyledon species (Appendix A). On the other hand, proximal duplication was detected for monocot species, namely *OsGSL1*, *OsGSL3*, *ScGSL5*, and *ScGSL6* (Appendix A). In apple, the WGD/segmental duplication is the major force responsible for the expansion of *MdGSLs* (Appendix A). Among 11 *PmGSLs*, 10 *PmGSLs* arose from dispersed duplication except for *PmGSL8*, which is likely generated from a WGD or segmental duplication event (Appendix A). We further performed positive selection tests on PmGSL protein residues and observed strong signs of purifying selection acting on most codons of GLS proteins with Ka/Ks ratios less than 1 across most protein residues (Appendix A).

### 2.6. Cis-Regulatory Element Analysis and Transcriptional Network Construction for PmGSL Genes

We examined the cis-regulatory elements within the promoter sequences of *PmGSL* genes and detected a number of hormone responsive cis-acting elements (Figure 4a; Appendix A). The ABA (abscisic acid) responsive element ABRE was abundantly present in most *PmGSL* promoters except for *PmGSL1*, *PmGSL8*, and *PmGSL10*. The TGA-element, which is responsive to auxin, was detected within the *PmGSL3*, *PmGSL4*, *PmGSL6*, *PmGSL8*, and *PmGSL11* promoters. Gibberellin responsive elements, namely the GARE-motif, P-box, and TATC-box were all absent from *PmGSL2*, *PmGSL5*, and *PmGSL6*. The MeJA responsive CGTCA-motif was highly enriched within the promoters of *PmGSL6* and *PmGSL11*, but was absent from the promoters of *PmGSL3* and *PmGSL5* (Figure 4a; Appendix A). Moreover, we identified a number of light-responsive cis-elements, including the AE-box, G-box, GA motif, CAG motif, GT1-motif, I-box, and GATA motif within most *PmGSL* promoters except for *PmGSL8*. The circadian element was only present in the *PmGSL8* promoter. The LTR (low-temperature responsiveness) element was only found in *PmGSL1*, *PmGSL3*, *PmGSL5*, *PmGSL6*, *PmGSL7*, and *PmGSL11*. Additionally, we detected the binding sites for MYB transcription factors, MYC transcription factors, and WRKY transcription factors among *PmGSLs* (Figure 4a; Appendix A).

We further inferred the potential transcription factors regulating *PmGSLs* by scanning the promoter sequences of *PmGSLs* using the online tool PlantRegMap (Figure 4b). The results showed that a total of 12 different transcription factor families were identified, including TCP (Teosinte Branched1/Cycloidea/PCF), bZIP, GRAS, and MADS-box (Figure 4b). Among the bZIP (Basic Leucine-Zipper) transcription factors, DREB2C (DRE-binding factor 2C) is a putative activator for *PmGSL2* and *PmGSL3* (Figure 4b). *PmGSL11* was predicted to be the common target gene for bZIP42, bZIP44, and bZIP53 in the network (Figure 4b). The MADS-box transcription factor SOC1 (Suppressor of Overexpression of Co 1) is predicted to regulate *PmGSL1*, *PmGSL2*, *PmGSL5*, *PmGSL6*, and *PmGSL11* (Figure 4b). RGA1 (Repressor of GA1), encoding a GRAS transcription factor, is possibly acting upstream regulating *PmGSL5*, *PmGSL6*, and *PmGSL11* (Figure 4b). We also detected a few transcription factors that were previously identified related to bud dormancy cycling. For example, ABI5 (ABA Insensitive 5), induced by ABA during bud endodormancy induction in *Pyrus pyrifolia* [28], was predicted to target *PmGSL5* in *P. mume*. Another bud endodormancy-associated TF, TCP20 possibly regulates *PmGSL9* along with TCP1 and TCP4 (Figure 4b) [29].

### 2.7. Tissue-Specific Expression Pattern Analysis of PmGSL Genes

To investigate the expression pattern of *PmGSL* genes across different tissues, we analyzed the FPKM values of *PmGSLs* across five organs of *P. mume* (Figure 5). Among all types of tissues, the stem has the highest number of *PmGSL* genes expressed, followed by the leaves and roots. We observed much higher transcription levels of *PmGSLs* in stem tissues than that in leaf tissues except for *PmGSL3* and *PmGSL10*. *PmGSL3* was found most highly expressed in leaves than in other tissues. *PmGSL10* was relatively highly expressed in fruits compared to other *PmGSL* genes. Interestingly, *PmGSL5* is the only gene member preferentially expressed in flower buds, while *PmGSL11* was specifically expressed in root tissues (Figure 5).

### 2.8. Expression Profile of GSL Genes during Floral Bud Development and Pathogen Infection

To explore the functional role of *GSL* genes in floral bud development, we analyzed the expression levels of *PmGSLs* during the flower bud break process in two *Prunus* species (Figure 6a,b). The *PmGSL* genes can be classified into three groups based on their expression profiles. *PmGSL4*, *PmGSL10*, and *PmGSL11* were relatively lowly expressed in endodormant floral buds, but were induced during floral bud flushing (Figure 6a). The transcript levels of *PmGSL2*, *PmGSL3*, and *PmGSL6* were relatively high in the endodormant floral bud tissues. After floral bud exit endodormancy, their expression levels dropped dramatically. The rest of the *PmGSLs* maintained high expression levels in floral bud tissues. However, their expression decreased in flushing flowers (Figure 6a). In *P. persica*, the *GSL* genes displayed two types of expression patterns during chilling accumulation in endodormant floral buds (Figure 6b). *PpGSL3*, *PpGSL4*, *PpGSL5*, and *PpGSL6* were significantly induced as floral buds accumulated chilling units, while the expression of *PpGSL1*, *PpGSL2*, *PpGSL7*, *PpGSL8,* and *PpGSL9* was decreased as floral buds exited endodormancy (Figure 6b).

To understand the role of *GSL* genes under biotic stress, we also analyzed the transcript levels of *PpGSLs* during fruit development in response to infection with *Monilinia laxa*, a fungal plant pathogen causing brown rot in many stone fruits [30]. In general, we observed two distinct expression patterns among *PpGSLs* (Figure 6c). *PpGSL2* and *PpGSL4* showed higher expression levels in mature fruits than in immature fruits. In healthy immature fruit, *PpGSL2* and *PpGSL4* were slightly down-regulated over time. Upon the pathogen infection, the expression of *PpGSL2* and *PpGSL4* was induced at 24 hpi (hours post inoculation) and 48 hpi, respectively (Figure 6c). In mature fruits, the expression of *PpGSL2* and *PpGSL4* first decreased over 24 h but peaked at 48 hpi under control conditions. With pathogen infection, *PpGSL2* and *PpGSL4* in mature fruits were significantly up-regulated at 24 hpi. Conversely, *PpGSL1*, *PpGSL3*, and *PpGSL5* showed relatively higher expression in immature fruits and were repressed after pathogen inoculation (Figure 6c). Similarly, *PpGSL6*, *PpGSL7*, *PpGSL8*, and *PpGSL9* were slightly repressed during fruit development after inoculation (Figure 6c).

### 2.9. The Hormonal Responsive Expression Pattern of PmGSL Genes

To further study the function of *PmGSLs* in hormonal signaling, we investigated the expression pattern of *PmGSL* genes in leaves treated with exogenous ABA, GA_3_, IAA, and MeJA using qRT-PCR assays (Figure 7). Upon the application of ABA, the expression levels of *PmGSL8*, *PmGSL10*, and *PmGSL11* were instantly up-regulated within first three hours, while the rest of the *PmGSLs* were slightly down-regulated within a few hours and were then slightly increased (Figure 7a). The expression of *PmGSL2*, *PmGSL3*, *PmGSL4*, *PmGSL8*, *PmGSL10*, and *PmGSL11* was first induced by IAA treatment and gradually increased until the sixth hour, then gradually declined. *PmGSL1*, *PmGSL5*, *PmGSL6*, and *PmGSL9* dropped slightly within the first hour; however, they again constantly increased reaching their maxima after six hours (Figure 7b). With the GA treatment, the expression profiles of most *PmGSLs* displayed similar trends compared with IAA treatment; while *PmGSL10* was strongly repressed in gibberellin-treated leaves (Figure 7c). We observed strong induction of most *PmGSLs*, including *PmGSL3*, *PmGSL4*, *PmGSL8*, *PmGSL10*, and *PmGSL11* after the application of MeJA. Their expression displayed slowly decreasing patterns after six hours (Figure 7d). Interestingly, *PmGSL7* was constantly decreased after all four types of hormonal treatments (Figure 7).

## 3. Discussion

Callose (β-1,3-glucan) is a linear polysaccharide that is widely produced in multicellular green algae and higher plants [31]. Callose is produced in specialized cell types at particular stages and plays an essential part in various biological processes associated with plant growth, development, and defense against biotic and abiotic stresses. During cell division, callose is transiently produced in the cell plate of phragmoplasts. Callose is also an important component of cell walls related to pollen development and self-incompatibility [32]. Callose deposition and degradation at plasmodesmata determines the intercellular transport of substances by controlling the pore sizes of sieve plates. When subjected to adverse environments, callose accumulates rapidly to seal the sieve pores, preventing nutrient loss of the source organs [33].

In plants, callose is synthesized primarily by callose synthase and is degraded into glucans by β-1,3-glucanase, also known as glycoside hydrolase family [33]. The callose synthase functions by forming a multi-subunit enzyme complex containing UDP-glucose transferase (UGT1) and sucrose synthase [34]. The UGT1 can interact with Rop1, the homologue of Rho protein, which regulates the activity of callose synthases [35]. In the best-studied model plant Arabidopsis, 12 callose synthase encoding genes have been identified and successfully cloned, named as *CalS1*-*CalS12* (*Callose synthase 1*-*Callose synthase 12*) or *GSL1*-*GSL12* (*Glucan synthase-like 1*-*Glucan synthase-like 12*) according to two nomenclatures. In this study, we adopted the more conservative GSL nomenclature for the callose synthase-encoding genes [32]. Previous studies revealed that *AtGSL1*, *AtGSL2*, *AtGSL5*, *AtGSL8,* and *AtGSL10* are mainly involved in pollen development and microsporogenesis [6,36]. Moreover, *AtGSL8* is responsible for the development of leaf stomata and cell plate formation during mitosis [14,37]. On the other hand, *AtGSL3*, *AtGSL4*, *AtGSL5*, *AtGSL7*, and *AtGSL12* are responsible for the callose deposition at plasmodesmata upon fungus infection, preventing the spread of pathogens [38]. In addition to *Arabidopsis*, the *GSL* genes were functionally characterized in other crop species such as rice [13] and maize [39].

Despite considerable advances in clarifying the function of GSL family genes in model plant species, their regulation and functional roles in tree species have been less explored. In poplar, callose deposition and hydrolysis at plasmodesmata was closely related to the closing and opening of transport channels for intercellular communication during bud dormancy cycling. Short photoperiods lead to elevated ABA levels and induce expression of SVP-like, which promotes the expression of *GSL* genes and the synthesis of callose at plasmodesmata in hybrid aspen [21,22]. During the dormancy maintenance, the plasmodesmata are blocked by callosic dormancy sphincters to prevent growth-promoting signals from reaching the apical meristem [40]. Subsequently, the endodormant buds accumulate sufficiently low temperatures, exit the dormancy state, and regrow, with callose degradation and plasmodesmata transport restored [41]. Similarly, callose deposition was detected in peach flower buds during flower organ differentiation and dormancy cycling [42]. In peach, the key dormancy regulator, PpDAM6 (Dormancy-associated MADS-box genes) can promote *PpCALS1* (*PpGSL2* in our study) and *PpCALS2* (*PpGSL7* in our study) gene expression and affect bud break by mediating plasmodesmata permeability [43]. During dormancy release, exposure to prolonged chilling periods increased GA levels and repressed the expression of *PpCALS1* and *PpCALS2* [43]. GSL family genes were reported to regulate pollen tube growth in pear [20] and pathogen defense responses to *Candidatus* Liberibacter asiaticus in citrus trees [44].

To obtain a deeper understanding of the GSL family genes in woody perennials, we systematically investigated the structural organization, evolution, and transcriptional regulation of *PmGSLs* genes. Moreover, we explored their possible transcriptional regulation and functional roles in *P. mume*. In the present study, we identified a total of 84 *GSL* genes from five eudicot species, namely *A. thaliana*, *Populus tricocarpa*, *M. domestica*, *P. mume*, and *P. persica*, and two monocot species, *O. sativa* and *S. bicolor*. The gene number of the GSL gene family is approximately even across species except for *M. domestica*. The two-fold gene number in *M. domestica* is likely due to whole-genome duplication which specifically occurred after the speciation of *Maleae* [45]. In *Arabidopsis*, the GSL family genes encode enzymes of about 2000 amino acids with extremely large molecular weights, making them difficult to analyze biochemically. In our analysis, the investigated GSL proteins exhibited a similar length and molecular weight up to 2009 aa and 230853.47 Da (MdGSL13). The gene structure analysis also showed large exon numbers of *PmGSLs* ranging from 19 to 47, which is similar to those of *AtGSLs* [6]. Usually, callose synthase contains a minimum of 10 transmembrane domains with a large central catalytic domain [6]. Our study confirmed that the PmGSL proteins were hydrophilic with 8 to 16 predicted transmembrane helices. Additionally, the subcellular localization of PmGSL proteins is predicted to be in the plasma membrane, which is consistent with previous findings that GSL proteins were found in membrane regions and vesicle-like structures in *Arabidopsis* [32].

The phylogenetic analysis of GSL family genes revealed six distinct gene clades, which is in agreement with previous findings [46]. For example, Yamaguchi et al. reported that *OsGSL8*, along with *OsGSL7* and *OsGSL11*, was grouped with *AtGSL3*, *AtGSL6*, *AtGSL9*, and *AtGSL12* as a sperate cluster [47]. Within each clade, the *GSL* genes from eudicot species were clearly separated from those of monocot species, indicating the common ancestral *GSL* genes underwent independent evolution after the split of these two lineages. The *PmGSLs* were grouped with *GSL* genes from *P. persica*, then with *M. domestica*, *Populus tricocarpa,* and *A. thaliana*, and finally with the monocotyledonous species *O. sativa* and *S. bicolor*. In general, the phylogeny among *GSL* genes clearly corresponds to the phylogenetic position of these species [46].

To understand the evolutionary trajectory of GSL family genes, we first studied the collinearity among GSL protein orthologues. We observed extensive syntenic relationships between the GSLs of two closely related species. However, no collinearity was observed between GSL orthologs of *Arabidopsis* and *O. sativa*. This is likely caused by extensive chromosome re-organization after the speciation of dicot and monocot species. The gene duplication analysis suggested that proximal duplication was mostly detected in monocot species. However, dispersed duplication, WGD/segmental duplication, and tandem duplication are the predominant duplication modes driving the expansion of GSL genes in dicotyledon species. In the positive selection test, we detected strong purifying selection acting among the residues within GSL proteins, which explains the sequence conservation observed within the functional domains of GSL proteins [46].

Previous studies have reported that plant hormones, such as auxin, abscisic acid, gibberellin, and salicylic acid are important in regulating the transcription of *GSL* genes [1]. For example, auxin regulates callose deposition and plasmodesma gating by activating the expression of *AtGSL8* to mediate the phototropic response in Arabidopsis [15]. To understand the transcriptional regulation of *GSL* genes in *Prunus mume*, we first analyzed the cis-regulatory element within the promoter of *PmGSLs* and detected abundant cis-acting elements related to response to various phytohormones and environmental stimuli, such as temperature and light. To further investigate the hormonal regulation of GSL gene transcription, we performed expression analysis and observed that all *PmGSLs* were either induced or repressed upon hormone treatment with abscisic acid, gibberellin, auxin, and MeJA. These results indicated that *PmGSL* genes are putatively regulated by transcriptional factors responsive to these phytohormones [1]. The RT-PCR assays showed that the expression of *PmGSL1*, *PmGSL3*, *PmGSL4*, *PmGSL9*, *PmGSL10*, and *PmGSL11* was responsive to exogenous gibberellin treatment. These results are consistent with the presence of GA-responsive elements within their promoter regions. Similarly, *PmGSL3*, *PmGSL4*, *PmGSL6*, *PmGSL8*, and *PmGSL11*, containing auxin-responsive elements, also displayed IAA inducibility. While the other *PmGSL* genes, lacking auxin-responsive elements within their promoters, also showed induced or repressed expression patterns within three hours after auxin treatment. Exogenous ABA was demonstrated to induce callose synthase activity and promote callose deposition in rice [48]. We observed the induced transcription of *PmGSL8*, *PmGSL10*, and *PmGSL11* upon application of abscisic acid, which suggested that these GSL genes may be involved in the abscisic acid-mediated signaling pathway. Previous studies also pointed out that jasmonic acid (JA) is required for proper callose deposition in pathogen-triggered plant defense responses [49]. In our study, we observed that *PmGSL3*, *PmGSL4*, *PmGSL8*, *PmGSL10*, and *PmGSL11* were induced after MeJA treatment, which suggests their possible role in defense against pathogens.

In addition to plant hormones, our analysis identified 31 transcription factors that possibly regulate the transcription of *PmGSL* genes. Among these TFs, DREB2C belongs to the dehydration-responsive element binding protein (DREB) family and is a regulator of plant responses to abiotic stresses such as drought and heat in Arabidopsis [50]. Our analysis predicts that *PmGSL2* and *PmGSL4* are targets of DERB2C, indicating these two genes are possibly involved in the abiotic stress-mediated callose deposition in *Prunus mume*. Moreover, a few predicted TFs were previously characterized in regulating bud dormancy in tree species. Previous studies have reported that dynamic accumulation of callose plays an important role in regulating the bud dormancy in vegetative buds and floral buds of tree species [21,22,42]. ABI5, a positive regulator of the ABA-signaling pathway, is found to promote bud dormancy in *Gladiolus* [51]. Its homologue, ABI3, was found crucial for short-day-mediated vegetative bud dormancy in poplar [52]. RGA1, a member of DELLA family, undergoes rapid degradation via 26S proteasome in response to gibberellin. RGL1 (RGA-LIKE 1), another DELLA member, negatively regulates chilling or gibberellin-induced dormancy release in endodormant buds of tree peony [53]. TCP20 is also involved in regulating floral bud endodormancy through inhibiting the expression of *PpDAM5/6* and *PpABF2* in peach [29]. The predicted target GSL genes, including *PmGSL5*, *PmGSL6*, *PmGSL9*, for these dormancy-related transcription factors may be essential for dormancy cycling in *Prunus mume*. By examining the expression pattern of GSL genes in two *Prunus* species, we observed three gene pairs, including *PmGSL6* and its orthologue *PpGSL2*, *PmGSL8* and its orthologue *PpGSL7*, and *PmGSL9* and its orthologue *PpGSL9*, which retained high expression levels in endodormant floral buds, but were down-regulated significantly as dormant buds accumulated chilling and exited dormancy. Based on previous findings, it is likely that these three *GSL* genes are involved in the dynamic callose deposition during the floral bud dormancy cycling in *Prunus* trees [43].

It is generally accepted that callose positively regulate plant defense responses to biotic and abiotic stresses. Various pathogenic viruses, bacteria, and fungi can trigger callose deposition at the sites of penetration [33]. Callose is specifically deposited between the plasma membrane and cell wall, and acts as a physical barrier to repress the spreading of pathogens. Typically, GSL family genes can be induced by conserved pathogen-associated molecular patterns (PAMPs), such as bacterial flagellin or fungal elicitor chitin, so as to promote callose deposition at plasmodesmata, restricting the plasmodesmata aperture and limiting the cell-to-cell spread of pathogens [54]. *AtGSL5* is responsible for the deposition of callose in papillae formation of Arabidopsis. A loss of function in *AtGSL5* displayed enhanced penetration of grass powdery mildew fungus on Arabidopsis [18]. In a previous report, immature peach fruit showed more resistance to *M. laxa,* with pathogen quiescence or death at 14–24 h after inoculation [30]. The transcription levels of *GSL* genes were much lower in immature fruits during the infection process compared to those in mature peach fruit, indicating the weak induction of *GLS* genes in more pathogen-resistant fruit tissues. We also observed strong up-regulation of *PpGSL2* and *PpGSL4* at 24 h after inoculation in mature fruits, which is likely caused by the increased pathogen activity after 6 h of infection [30]. The up-regulation of *PpGSL2* and *PpGSL4* suggested that they are mainly involved in the callose deposition during the early defense mechanism against pathogen attacks in *P. persica*.

## 4. Materials and Methods

### 4.1. Identification of GSL Family Genes in Seven Plant Species

The up-to-date genomes of *A. thaliana*, *Populus tricocarpa*, *M. domestica*, *P. mume*, *P. persica*, and two monocot species, *O. sativa* and *S. bicolor*, were downloaded from the public datasets TAIR (http://www.arabidopsis.org, accessed on 1 August 2023) and Phytozome (https://phytozome-next.jgi.doe.gov, accessed on 1 August 2023). To identify the *GSL* gene members for each plant species, we first searched all plant genomes for putative GSL proteins using HMMER 3.1 software based on the hidden Markov model for glucan synthase (PF02364) (https://www.ebi.ac.uk/Tools/hmmer/, accessed on 1 August 2023). Using the protein sequences of Arabidopsis GSL proteins as query, we blasted against the genomes of six other species and obtained putative GSL proteins with e-values ≤ 1.0 × 10^−10^. Furthermore, we combined all candidate *GSL* genes identified using two approaches and confirmed all proteins belonging to the glucan synthase superfamily using NCBI CDD search (https://www.ncbi.nlm.nih.gov/Structure/cdd/wrpsb.cgi, accessed on 1 August 2023) and NCBI BLAST (https://blast.ncbi.nlm.nih.gov/Blast.cgi, accessed on 1 August 2023). Finally, the non-redundant genes with complete GSL domains were used in the following analysis.

### 4.2. Sequence Analysis and Structural Characterization of PmGSL Genes

We mapped all *PmGSL* genes to chromosomes and retrieved their gene annotation information. The gene structures and chromosomal locations of *PmGSL* genes were visualized using TBtools software v2.028 [55]. All genes were renamed in order according to their positions among chromosomes. We analyzed the protein sequences of PmGSL proteins by extracting the amino acid sequences within the GSL featured domains and aligned them with MUSCLE software v5.0 [56]. Multiple alignment was visualized using GeneDoc v2.6 [57]. Conserved protein motifs were also predicted for all PmGSL proteins using MEME (http://meme-suite. org/tools/meme, accessed on 10 August 2023). We further assessed the protein physicochemical characteristics for all PmGSL proteins including the isoelectric point, number of amino acids, and molecular weight using the ExPASy online tool (http://web.expasy.org/protparam/, accessed on 10 August 2023). The protein transmembrane helices were predicted using TMHMM server v2.0 (http://www.cbs.dtu.dk/services/TMHMM-2.0/, accessed on 10 August 2023). The protein subcellular locations were also predicted for *PmGSL* genes using Plant-mPLoc program (http://www.csbio.sjtu.edu.cn/cgi-bin/PlantmPLoc.cgi, accessed on 10 August 2023).

### 4.3. Phylogenetic Analysis of GSL Family Genes

To understand the phylogeny among the GSL proteins from seven plant species, the whole sequences of GSL proteins were aligned using the software MUSCLE v5.0. Based on the multiple sequence alignment, the phylogenetic tree of all GSL family proteins was constructed with FastTree using the maximum likelihood (ML) method [58]. The phylogenetic tree of PmGSL proteins was built using the same approach. Finally, the phylogenetic tree of all GSL proteins was visualized using the online tool iTOL (http://itol2.embl.de, accessed on 10 August 2023). To study the sequence conservation within GSL functional domains, we also extracted the amino acids of all PmGSLs within GSL domains and visualized the sequence alignment using GeneDoc v2.6.

### 4.4. Microsynteny and Selection Analysis of GSL Family Genes

To infer the synteny among GSL family members among species, we performed all-against-all BLASTP among genomes of seven investigated plant species. The software Multiple Collinearity Scan toolkit X version (MCScanX) was used to analyzed the interspecific synteny relationship among *GSL* gene pairs across species based on the top BLASTP hits (e-values < 1.0 × 10^−10^) [59]. The intraspecific synteny relationship was analyzed to identify the duplication events of *PmGSL* family genes using the ‘duplicate_gene_classifier’ in MCScanX. The intra-species and inter-species synteny among GSL family genes was visualized using the software TBtools v2.028. Moreover, we also analyzed the mode of selection acting among codons of PmGSL proteins using the Selecton server (http://selecton.tau.ac.il; accessed on 6 August 2023).

### 4.5. Promoter Analysis and Transcriptional Regulator Prediction of PmGSLs

We extracted the promoter sequences of *PmGSL* genes as the 2000 bp upstream genomic sequences for each gene. The cis-acting elements were predicted with the PlantCARE database (http://bioinformatics.psb.ugent.be/webtools/plantcare/html/, accessed on 10 August 2023). To screen for the putative transcription factors (TFs) regulating *PmGSL* genes, we used the Plant Transcriptional Regulatory Map (PTRM) tool (http://plantregmap.gao-lab.org/regulation_prediction.php, accessed on 10 August 2023) for the prediction of TFs that possibly bind to *PmGSL* genes at the threshold parameter *p*-value ≤ 1 × 10^−6^. The predicted TF-gene regulatory network was further visualized using the Cytoscape software v3.10.0 [60].

### 4.6. Expression Pattern Analysis of PmGSL Genes

The expression patterns of *PmGSL* genes across different plant tissues were analyzed using the previously published transcriptome data of *P. mume* (GSE4760162) from NCBI SRA database. To investigate the role of *GSL* genes during flower bud dormancy, we obtained the transcriptome data of *P. mume* [61] and *P. persica* (GSE189882) [42]. Moreover, we collected the RNA-seq data for the infection of *Monilinia laxa* during fruit development of *P. persica* (GSE146293) [30]. For all transcriptome data, raw sequencing reads were preprocessed using fastp [62] and were aligned to their corresponding reference genomes using HISAT2 [63]. The FPKM (transcript reads per million mapped reads) values were extracted for GSL genes and were compared across different tissues or developmental stages using the R package ‘edgeR’. Finally, the heatmaps of *GSL* genes were generated using the ‘pheatmap’ package in R.

### 4.7. Plant Hormone Treatment and Quantitative Real-Time Polymerase Chain Reaction

To study the expression profiles of *GLS* genes in response to different hormones, we applied different hormones including ABA (CAS#: BN20101, Biorigin Inc., Beijing, China), MeJA (CAS#: 392707, Sigma-Aldrich Inc., St. Louis, MO, USA), IAA (CAS#: BN20265, Biorigin Inc., China), and GA_3_ (CAS#: BN20126, Biorigin Inc., China) on leaves of a five-year old *P. mume* tree. The fully expanded leaves were treated with 100 mM IAA, 100 mM GA_3_, 2 mM MeJA, or 300 mg/L ABA. We collected leaves from current-year branches at 0, 1, 3, 6, 12, 24, and 48 h after hormone treatment and froze the samples immediately with liquid nitrogen.

The total RNA was extracted from samples with the Plant RNA Kit (Omega Bio-tek, Norcross, GA, USA) following the protocol. The quality and concentration of RNA was measured with a Thermo Scientific NanoDrop Spectrophotometer (Thermo Scientific, Waltham, MA, USA). The total RNA was used for cDNA synthesis using the PrimeScript RT reagent kit with gDNA eraser (Takara, Shiga, Japan). We performed qRT-PCR assays with three technical replicates on the PikoReal real-time PCR platform (Thermo Fisher Scientific, Dreieich, Germany) to assess the expression levels of *GSL* genes. The temperature settings of qRT-PCR experiments were as follows: 95 °C for 30 s; 40 cycles of 95 °C for 5 s, 60 °C for 30 s; 72 °C for 30 s. The relative gene expression levels were calculated with protein phosphatase 2A (PP2A) gene as the internal reference using the 2^−ΔΔCt^ method. The primers used for qRT-PCR experiments are provided (Appendix A).

## 5. Conclusions

In this study, we systematically identified and explored the gene structural organization, evolutionary trajectory, transcriptional regulation, and molecular function of GSL family genes in *P. mume*. The protein sequence alignment revealed strong conservation among residues within GSL functional domains. However, the exon–intron structure and protein properties varied slightly among *PmGSL* family genes. With evolutionary analysis, we observed different types of duplication events and strong purifying selection contributing to the expansion and evolution of GSL family genes. Moreover, we analyzed the cis-regulatory elements within gene promoters and inferred the transcription factors putatively regulating *PmGSL* genes. By examining the gene expression pattern across different tissues and during diverse biological processes, we identified a number of *GSL* genes essential for floral bud dormancy cycling, hormonal responses, and pathogen infection in *Prunus* species. In summary, the findings of this study can facilitate more in-depth understanding of callose synthase genes in *P. mume* and have laid the foundation for future functional research of glucan synthase genes in other perennial tree species.

## Figures and Tables

**Figure 1 plants-12-04159-f001:**
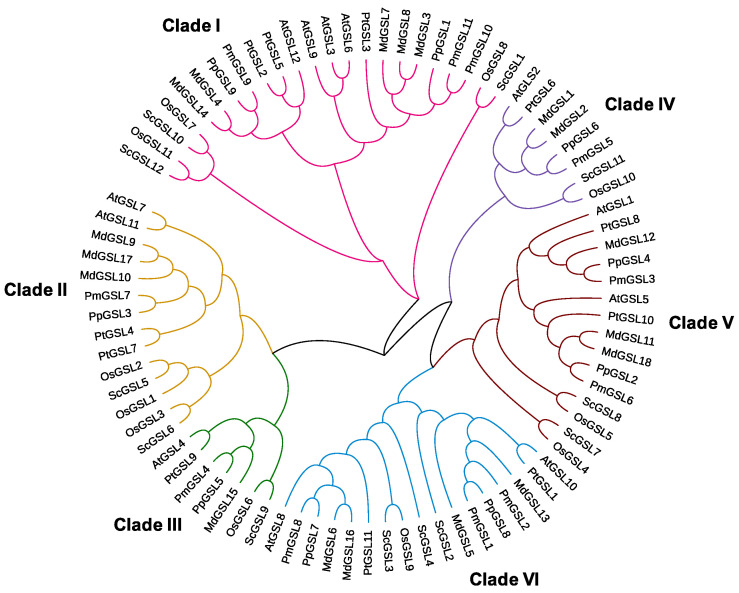
The phylogenetic tree of 84 GSL genes from seven plant species based on their protein sequence alignment using a maximum likelihood approach. The GSL genes were clustered into six clades colored differently.

**Figure 2 plants-12-04159-f002:**
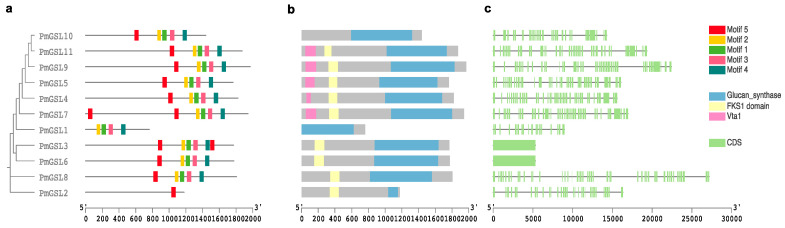
Gene structure and motif analysis of *PmGSL* genes. (**a**) The phylogenetic tree of PmGSL protein sequences was built with protein motifs colored differently. (**b**) The functional domains characterized among PmGSL proteins using the NCBI CDD tool. (**c**) The exon–intron distribution of *PmGSL* genes. The green boxes represent exons and the black lines represent intron positions.

**Figure 3 plants-12-04159-f003:**
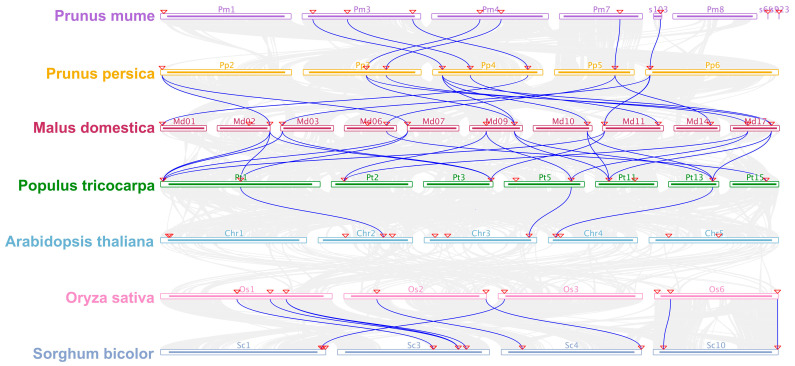
The genome synteny analysis of *GSL* genes from *A. thaliana*, *M. domestica*, *P. mume*, *P. persica*, *Populus tricocarpa*, *S. bicolor*, and *O. sativa.* The red triangles highlight the GSL family genes and the blue lines connect syntenic *GSL* gene pairs. The grey shades represent the syntenic regions across genomes.

**Figure 4 plants-12-04159-f004:**
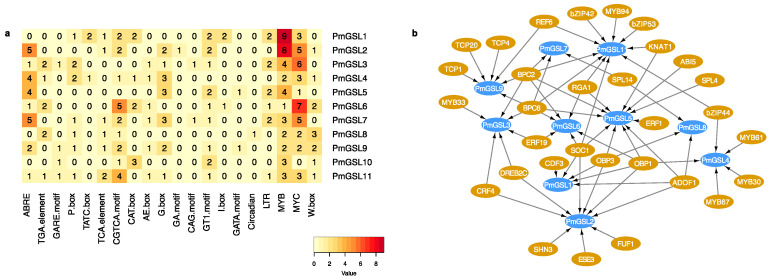
The promoter sequence analysis and regulatory factor prediction for *PmGSL* genes. (**a**) The cis-regulatory elements identified within *PmGSL* gene promoters. The numerical values and colors represent the number of cis-elements identified within each *PmGSL* gene. (**b**) The transcriptional regulatory network of *PmGSL* genes inferred based on the Plant Transcriptional Regulatory Map (PTRM) database. The arrows indicate putative transcriptional regulation relationships between *PmGSL* genes and their upstream TFs.

**Figure 5 plants-12-04159-f005:**
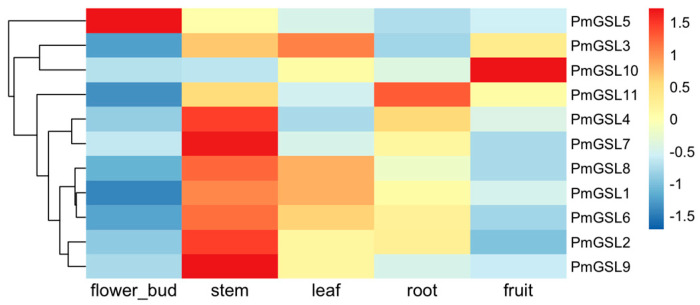
The tissue-specific expression profile of *PmGSL* genes in *P. mume*.

**Figure 6 plants-12-04159-f006:**
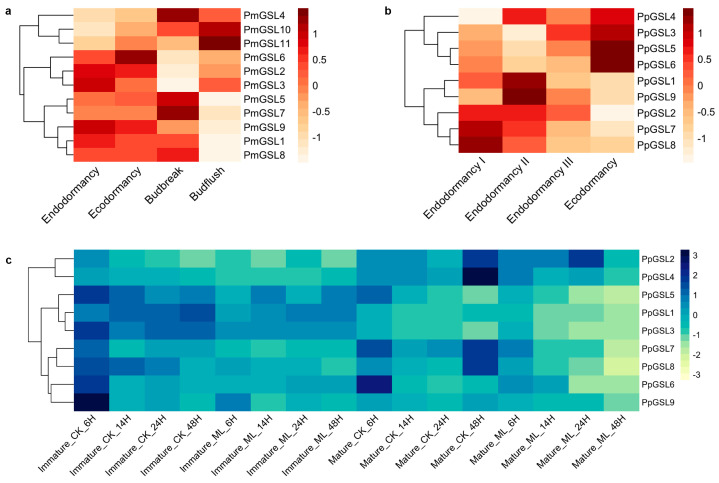
The expression profile of *PmGSLs* during flower bud dormancy release and pathogen infection in *Prunus* species. (**a**) The expression pattern of floral bud break in *P. mume*. (**b**,**c**) The expression pattern of *PpGSLs* during flower bud dormancy release (**b**) and fruit infection with *Monilinia laxa* (**c**) in *P. persica*.

**Figure 7 plants-12-04159-f007:**
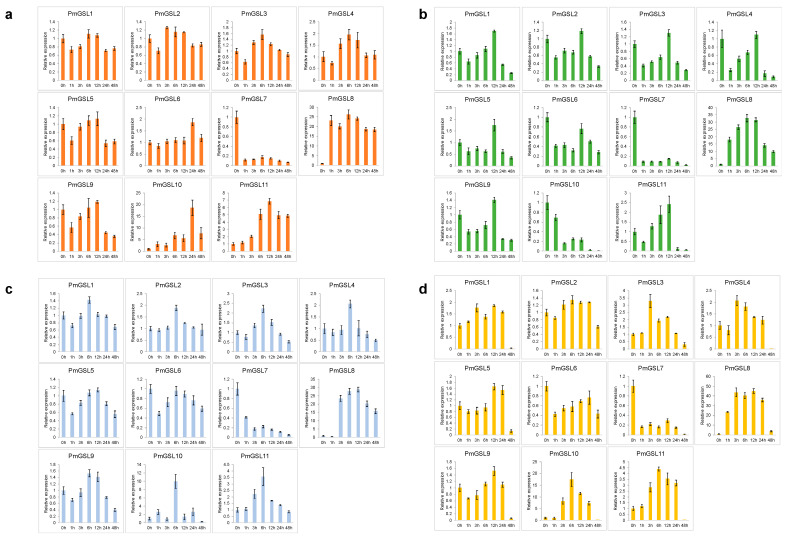
The relative expression levels of *PmGSLs* in leaves treated with the following hormonal treatments, including ABA (**a**), IAA (**b**), GA_3_ (**c**), and MeJA (**d**) treatments. The error bars indicate the standard deviation of the relative expression values calculated for each gene across time points.

**Table 1 plants-12-04159-t001:** The protein physiochemical analysis of GSL family genes identified in *Prunus mume*.

Gene Name	Protein Length	MW/Da	pI	Instability Index	Aliphatic Index	GRAVY	Number of Transmembrane Helices	Subcellular Prediction
PmGSL1	761	85,842.58	8.89	22.08	100.7	0.209	8	Plasma membrane
PmGSL2	1177	136,609.85	8.04	42.56	95.18	−0.174	6	Plasma membrane
PmGSL3	1769	204,837.86	9.03	40.86	100.22	0.034	16	Plasma membrane
PmGSL4	1823	210,775.28	8.61	45.53	92.78	−0.096	10	Plasma membrane
PmGSL5	1763	202,350.42	8.79	42.42	97.27	−0.069	12	Plasma membrane
PmGSL6	1773	206,642.53	9.0	37.91	97.16	−0.017	16	Plasma membrane
PmGSL7	1943	225,407.81	8.79	39.16	92.95	−0.097	14	Plasma membrane
PmGSL8	1805	205,401.5	8.47	37.78	94.6	−0.068	12	Plasma membrane
PmGSL9	1970	227,615.31	9.13	45.42	93.28	−0.1	14	Plasma membrane
PmGSL10	1438	165,118.44	9.05	36.42	93.88	−0.054	12	Plasma membrane
PmGSL11	1873	215,381.89	8.96	39.76	89.11	−0.167	14	Plasma membrane

## Data Availability

The datasets used in this study are publicly available. All analyzed data can be found in the article or in the Appendix A.

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
