# Peer review of "Genome-Wide Identification of Callose Synthase Family Genes and Their Expression Analysis in Floral Bud Development and Hormonal Responses in Prunus mume"

_plants, 2023, doi:10.3390/plants12244159_

Round 1
Reviewer 1 Report
Comments and Suggestions for Authors
In the manuscript entitled 'Genome-wide identification of callose synthase family genes 2 and their expression analysis in floral bud development and 3 hormonal responses in Prunus mume', the authors mainly used bioinformatic analysis to illustrate the gene evolution, structure and the responsiveness to infections of glucan synthase like (GSL) enzymes and their protein properties regarding MW, PI, transmembrane domains, and subcellular location. etc In addition, the authors did the analysis of the gene expression with treatment of plant hormones like ABA, MeJA, IAA and GA3. Overall the writing and the structure is good. The topic and content may be interesting for readers from different ears including agriculture, plant protection. Some specific comments and questions as below:
1) What are the reasons why the authors chose the specific plant hormones but not others and what are the conclusion and implication? What are the error bars?
2) Could the authors comment whether all the pmGSL member gene can be translated into protein and explain?
3) There is no information of the chemicals used in the work in the Section 4 Materials and Methods.
Author Response
Dear reviewer, Thank you very much for providing comments and suggestions on our manuscript. We have studied your comments carefully and have revised our manuscript accordingly. Attached please find the revised version and response letter, which we would like to submit for your kind consideration.
Great thanks,

Reviewer 2 Report
Comments and Suggestions for Authors
The following comments need to be addressed by the authors in the revised submission.
1. Gene names should be in italics throughout the manuscript
2. In the manuscript tables should be given in increasing order (Table S2 has been given before Table S1 in the manuscript)
3. In the Table S2 include chromosome information of the GSL genes as well as their Subcellular location information identified in different plant species
4. In Table 1 chromosome information should be included
5. Adjust the font size in figure 2a to visible gene names more clearly and improve the visibility of figure 2.
6. In line no. 188-189, It is mentioned that chromosome 8, and three scaffolds in the form of single gene copy while in the figure S3 there is no PmGSL gene present in chromosome 8.
7. Improve the quality of figure 3
8. Use short form of the species names while using more than once in the manuscript (eg. Populus tricocarpa should be written as P. trichocarpa)
9. Increase the font size in figure 7 to improve the visibility of text
10. Performed GO enrichment analysis to understand the function of predicted GSL family genes.
Comments on the Quality of English LanguageEnglish language needs attention.
Author Response

(The authors gave the same response as above.)

Reviewer 3 Report
Comments and Suggestions for Authors
Minor editing of the English language required
Author Response
Dear reviewer, Thank you very much for providing comments and suggestions on our manuscript. We have studied your comments carefully and have revised our manuscript accordingly. Attached please find the revised version and response letter, which we would like to submit for your kind consideration.
Great thanks and Best regards !
